# Effectiveness of Health Coaching in Smoking Cessation and Promoting the Use of Oral Smoking Cessation Drugs in Patients with Type 2 Diabetes: A Randomized Controlled Trial

**DOI:** 10.3390/ijerph20064994

**Published:** 2023-03-12

**Authors:** Li-Chi Huang, Yao-Tsung Chang, Ching-Ling Lin, Ruey-Yu Chen, Chyi-Huey Bai

**Affiliations:** 1Endocrinology & Metabolism, Cathay General Hospital, Taipei 106438, Taiwan; 2School of Public Health, Taipei Medical University, Taipei 110301, Taiwan; 3School of Medicine, College of Medicine, Taipei Medical University, Taipei 110301, Taiwan; 4School of Medicine, National Tsing Hua University, Hsinchu 300044, Taiwan

**Keywords:** health coaching, smoking cessation, smoking reduction, cigarette, type 2 diabetes, motivational interviewing

## Abstract

Introduction: This study looked into the effectiveness of a 6 month health coaching intervention in smoking cessation and smoking reduction for patients with type 2 diabetes. Methods: The study was carried out via a two-armed, double-blind, randomized-controlled trial with 68 participants at a medical center in Taiwan. The intervention group received health coaching for 6 months, while the control group only received usual smoking cessation services; some patients in both groups participated in a pharmacotherapy plan. The health coaching intervention is a patient-centered approach to disease management which focuses on changing their actual behaviors. By targeting on achieving effective adult learning cycles, health coaching aims to help patients to establish new behavior patterns and habits. Results: In this study, the intervention group had significantly more participants who reduced their level of cigarette smoking by at least 50% than the control group (*p* = 0.030). Moreover, patients participating in the pharmacotherapy plan in the coaching intervention group had a significant effect on smoking cessation (*p* = 0.011), but it was insignificant in the control group. Conclusions: Health coaching can be an effective approach to assisting patients with type 2 diabetes participating in a pharmacotherapy plan to reduce smoking and may help those who participate in pharmacotherapy plan to quit smoking more effectively. Further studies with higher-quality evidence on the effectiveness of health coaching in smoking cessation and the use of oral smoking cessation drugs in patients with type 2 diabetes are needed.

## 1. Introduction

Diabetes is a chronic disease which inflicts heavy burdens on medical systems and social systems, as well as causes numerous deaths, and the number of people throughout the world suffering from diabetes is increasing. The World Health Organization (WHO) notes that the global prevalence rate of diabetes among adults now tops 8.5% [1]. In Taiwan, the prevalence rate of diabetes among the population aged 18 and older stands at around 9.82% [2], ranking second place in health insurance outlay [3]. The prevention and treatment of diabetes involve lifestyle modification, among which smoking cessation is an important and challenging task to address. Smoking can cause insulin resistance, thereby increasing the risk of diabetes and accelerating the rate of diabetic deterioration [4]. Although nicotine withdrawal may increase the risk of obesity due to an increased appetite in the short term, it gradually decreases as the time from quitting smoking increases [5]. In the long run, quitting smoking has many benefits for people with diabetes. However, the association between smoking and diabetes is often overlooked by patients.

The smoking rate in Taiwan in 2020 was about 13.1%, of which men aged 31–50 had the highest prevalence of smoking, reaching a high of 30%. Since 2012, Taiwan has promoted “The second-generation smoking cessation program”, which has greatly extended drug subsidy and increased participation flexibility [6]. According to the Health Promotion Administration (HPA) Taiwan, the 6 month smoking cessation clinical services, which combine the smoking cessation drugs and counseling services, can achieve a 27.1% smoking abstinence rate, and the free online smoking cessation consultation hotline also has a 39.5% smoking abstinence rate [6]. There are more studies conducted on groups with higher smoking rates, such as males and disadvantaged groups, or groups of patients who are more easily affected by smoking behaviors, such as pregnant women, hypertensive persons, and lung cancer patients, with relatively few studies focusing on people with diabetes [7,8,9]. Although there are some studies indicating the benefits of smoking cessation to patients with diabetes [10], in Taiwan, the current evidence of smoking cessation in diabetic patients appears to be insufficient.

There are quite a few studies on smoking cessation interventions using health education or behavioral counseling for more than 20 years, and motivational interviewing (MI) is one of the techniques used to aid quitting [7,11]. It involves a series of guided questions specifically aimed at patients’ motivation to change, especially focusing on inducing ambivalence to increase patients’ motivation. However, in recent years, health coaching has emerged as an increasingly popular approach to behavioral change; however, since there seems to be only a very small number of smoking cessation-related health coaching studies, the behavioral counseling for smoking cessation is still dominated by MI. Health coaching is a patient-centered approach to disease management which focuses on patients’ decisions and their actual behaviors [12]; it is based on positive psychology and humanistic psychology, integrating many behavioral counseling theories and techniques such as MI, appreciation inquiry, design thinking, solution-focused therapy, and adult learning theory, and it focuses on achieving effective adult learning cycles to enable patients to make changes and establish new behavior patterns and habits. Compared with MI, health coaching is a more comprehensive approach, which makes up for the limitations of using MI alone, as it utilizes various skills and is more focused on adult learning and empathy. In recent years, in addition to traditional coach training institutions (such as the International Coach Federation, ICF), there are some institutions such as the National Board for Health and Wellness Coaching (NBHWC) that specialize in certifying and training health coaches and have a clear definition and specification for the coach [13,14]. Therefore, health coaching has a more complete framework than counseling using MI or any other technique alone, and there are international training, certification, and monitoring bodies that clearly define coaching and its practice [14]. Although no studies are available to compare whether implementing integrative method such as health coaching is significantly better than using a single technique such as MI, health coaching has become a mainstream in behavior modification for patients with chronic disease such as diabetes mellitus in recent years.

However, because many health coaching studies often lack sufficient information on coach training, certification, and supervision, or do not clearly describe the methods used by coaches, it is impossible to effectively confirm whether coaches truly meet the definition of coaching by the aforementioned institutions, and many existing coaching studies may still be health education or counseling [12,15,16]. Therefore, the aim of this study was to test the effectiveness of a health coaching intervention to help diabetic patients quit smoking and/or participate in a pharmacotherapy plan under the premise of providing sufficient intervener information.

## 2. Materials and Methods

The study involved a 6 month coaching intervention in a two-armed, double-blind, randomized controlled trial, approved by the Institutional Review Board (IRB) of Cathay General Hospital (Taipei, Taiwan). All relevant ethical safeguards were met in relation to participant protection, and ethical standards were in accordance with the Declaration of Helsinki. Two groups of subjects participated: one with a monthly coaching intervention and the other with usual care only.

### 2.1. Study Procedures and Randomization Settings

Participants were recruited among patients with diabetes treated at the Department of Endocrinology and Metabolism at Cathay General Hospital (Taipei, Taiwan) from August 2020 and May 2021; however, this study had to be discontinued in May 2021 due to the coronavirus disease 2019 (COVID-19) pandemic. In addition, unfortunately, due to the risk from nitrosamine impurity in varenicline, it was requested to be taken off the shelves by the health authorities in Taiwan in July 2021 [17], and all the smoking cessation projects in Taiwan that year were suspended until the following year. The suspension of cessation programs along with the drug recall halted patients’ participation in smoking cessation drug treatment, resulting in the absence of post-test data collection. A diabetes health educator screened and tagged patients with type 2 diabetes and smoking habit from the hospital’s database, followed by an independent researcher who randomly assigned them to the intervention group and the control group using computer-generated random numbers by the PASW 22.0 software for windows (SPSS, Chicago, IL, USA). Then, two physicians recruited them individually. Patients in the intervention group were informed of the coaching program by a health coach, whereas the control group was provided with the usual smoking cessation plan (Figure 1). Therefore, the physicians, patients, and analysts were blinded, the participants in the control group did not know that this was an interventional trial, and the coach was not blinded since he was aware of the patients in the intervention group.

The inclusion criteria were as follows: age 20 to 75 years with type 2 diabetes for at least 1 year, Mandarin Chinese or Taiwanese as the spoken language, smoking one or more cigarettes per day, and not taking cessation medication or joining any cessation plan. The exclusion criteria were signs of clinical depression or cognitive function impairment.

Between August 2020 and May 2021, prior to the COVID-19 restrictions in May 2021 in Taiwan, 92 potential subjects were screened from database but only 74 subjects were enrolled in the study. In total, outcome measures were available for 35 participants in the intervention group and 33 in the control group, while six participants withdrew from the follow-up due to the drug recall (Figure 1). The recruitment rate was 80.4%, and the main reason for participants’ refusal was the unwillingness to quit smoking at that stage.

### 2.2. Pharmacotherapy Plan

Both groups could freely choose to participate in a 2 month varenicline treatment plan. In Taiwan, the National Health Insurance (NHI) provides a maximum of two varenicline treatment subsidies a year, each for a period of 2 months, with three-stage drug treatment [18]. Only those assessed with the Fagerström Test for Nicotine Dependence (FTND) with a score of >3 could participate in the program. Participants had to return to the clinic in the third and seventh weeks to receive medicine, and their nicotine addiction was assessed again. The degree of addiction was tracked in the 12th and 24th weeks. Participants of the intervention group were referred to the case manager after participating in the pharmacotherapy plan. During the 2 month varenicline treatment, they also received coaching from the coach when returning to the hospital; afterward, coaching intervention resumed until the end of the 6 month intervention. Therefore, participants in the intervention group were tracked by the coach after the pharmacotherapy plan, while those in the control group were tracked by the case manager only. In total, four case managers were responsible for managing and tracking all the patients in the pharmacotherapy plan, and those four were nurses who are qualified diabetes health educators.

### 2.3. Intervention

Patients in the intervention group received in-person coaching at baseline and were followed up by a monthly telephone call for a total of 6 months, or they could participate in the pharmacotherapy plan and have a coaching intervention the other 4 months. Coaching was provided on a one-on-one basis by a professional health coach who has a master’s degree in public health and received over 120 h of coach training before being certified as an International Coach Federation (ICF)’s Associated Certified Coach (ACC).

To be specific, the coach used techniques such as MIs, appreciation inquiry, and value exploration to arouse participants’ motivation to quit smoking and allowed participants to weigh the pros and cons of quitting smoking due to diabetes and the needs of their personal work or family life. The coach used the patient-centered principle, self-determination theory, and transtheoretical model to guide the patients to think via techniques such as active listening and open-ended questions, instead of using traditional preaching methods of health education, thereby strengthening the psychological support and motivation felt by the patients. For some patients who have irrational beliefs about smoking behavior, such as the belief that smoking is a necessary means to establish working partnerships or carry out their creative work, the coach tried to use rational emotive cognitive behavior coaching (RECBC) skill for coaching.

In the first session, the coach asked each participant to establish their 6 month smoking cessation goals, and then the coach assisted them in formulating a plan to gradually reduce smoking. Participants needed to design “SMART” (i.e., specific, measurable, attainable, realistic, and timely) goals for their action plans, and they could design a smoking reduction plan on a weekly or monthly basis, or just stop smoking straight away. In addition, the coach also asked the participants if they felt the need to use smoking cessation drugs, and some decided to participate in the pharmacotherapy plan during or after the initial coaching session. Most coaching calls were made while the participants were at home or during lunch break. During follow-up telephone coaching sessions, the coach continued to enhance each patient’s motivation to quit smoking, reviewed the goal and action plan with the patient, and discussed possible obstacles to quitting and the solutions together. Through coaching, each patient could rethink the necessity of smoking and its health effects; however, even if the patient chose to just reduce the number of cigarettes instead of quitting smoking entirely due to work or family needs, the coach still respected the patient’s choice. In general, it took more time, about 30–45 min, in the first coaching session than in follow-up telephone coaching sessions.

Both intervention and control group patients received the usual smoking cessation education and tracking. Currently, the smoking cessation health education adopted in Taiwan includes the five As (ask, advise, assess interest, assist, and arrange) and five Rs (relevance, risks, rewards, roadblocks, and repetition) strategy of smoking cessation promoted by the WHO to strengthen the effectiveness of smoking cessation [19]. Health educator also provides health education relating to smoking cessation such as the disadvantages of smoking, withdrawal syndromes, and medication information according to the standardized smoking cessation consulting training designed by the Health Promotion Administration, Ministry of Health and Welfare Taiwan [20]. Patients in the control group did not receive any additional intervention and only had follow-up calls from the case manager in the 4th, 8th, 12th, and 24th weeks, where they were asked if they had quit smoking and/or required any health education or treatment needs. Therefore, patients in the control group had fewer follow-up times than those in the intervention group.

### 2.4. Sample Size

Considering that smoking cessation behaviors may be affected by the current policy environment and culture in Taiwan, when calculating the sample size, we referred to the sample size calculation of medical or behavior intervention studies conducted in Taiwan and published in international peer-reviewed journals in recent years [21]. It seems that if effect size is used as a sample size estimation, a moderate effect size (ES = 0.5) may be an appropriate estimation criterion. Hence, to detect a 0.50 effect size with a probability of a type I error of 0.05 and a power of 80%, each group required at least 64 participants.

### 2.5. Outcome Measures

The main outcome variables of this study were smoking abstinence and smoking reduction. The smoking abstinence was assessed by Fagerström Test for Nicotine Dependence (FTND), with zero points indicating success in quitting smoking. Although there are many smoking cessation studies using carbon monoxide measurement as the smoking cessation assessment indicator to determine whether a patient has smoked during the assessment period [22], it was not used in this study due to the COVID-19 epidemic prevention considerations; instead, the FTND was used. This self-assessment tool has six items with an overall score ranging 0–10 [23], and it is also the main smoking abstinence criterion for the second-generation smoking cessation service in Taiwan. The smoking reduction was assessed by participants’ self-reported number of cigarettes smoked per day, and a significant reduction was defined as they reduced their daily smoking by at least 50% from baseline [24]. The independent variables included the coaching intervention, baseline number of daily cigarettes, length of smoking (years), the FTND score, pharmacotherapy plan participation, experience with smoking cessation, and sociodemographic characteristics. Sociodemographic characteristics included gender, age, educational level, and job position.

### 2.6. Statistical Analysis

A chi-square test or *t*-test was employed to assess differences in sociodemographic variables and baseline smoking behaviors. The chi-square test was used to assess the difference in smoking cessation and reduction in smoking at the follow-up, and the paired *t*-test was used to assess the reduction in the number of cigarettes. The effect size of intervention was also calculated. Lastly, we used multivariate logistic regression to test whether coach intervention has higher odds of quitting smoking than the usual smoking cessation plan in Taiwan and adjusted the baseline value of daily cigarettes. Since there were no significant differences in demographic characteristics between the two groups, we included only the daily number of cigarettes at baseline as an adjustment variable.

All tests were analyzed at a 95% significance level (*p* < 0.05). Intention-to-treat analysis was not used since the ethical policy states that noncompliers who refuse to continue to participate are to be excluded from the analysis. Analyses were conducted using PASW 22.0 software for windows (SPSS, Chicago, IL, USA).

## 3. Results

Table 1 shows the demographic characteristics of the 68 participants: 82.4% were male, the mean age was 56.0 (standard deviation (SD) = 10.62) years, 39.7% had a bachelor’s degree or higher, the mean length of smoking was 34.5 (SD = 9.70) years, mean addiction to nicotine was 5.3 (FTND score, SD = 2.46) points, 82.4% had never tried to quit smoking, and about half of the participants were willing to use varenicline to quit smoking. There were no significant demographic differences between the two groups at baseline.

With the 6 month coaching intervention or the 2 month varenicline use plus 4 month coaching intervention, the intervention group and the control group had 48.6% and 36.4% of participants, respectively, who quit smoking for a nonsignificant difference in the smoking cessation rate between the two groups (*p* = 0.309, Table 2). However, the intervention group had significantly more participants who reduced their smoking than the control group (*p* = 0.030), and there was a significantly greater reduction in the number of cigarettes than the control group (*p* = 0.032) with Cohen’s d of 0.53. In the intervention group, 54.3% of participants decreased their smoking with a significant (*p* < 0.001) decrease of 12.88 (SD = 9.28) cigarettes per day. In the control group, 21.2% of participants decreased their smoking with a significant (*p* < 0.001) decrease of 7.74 (SD = 10.03) cigarettes per day.

Health coaching intervention seemed to improve the effectiveness of smoking cessation among varenicline users and smoking reduction for the participants without receiving varenicline treatment (Table 3). With the use of varenicline, it seems that the intervention group had more participants quit smoking than the control group (*p* = 0.082). In contrast, there were more participants in the intervention group reducing the number of cigarettes smoked when receiving coaching only rather than with the use of varenicline (*p* = 0.014).

As to the smoking cessation rate, the multivariate logistic regression revealed that only varenicline use (odds ratio (OR) = 3.67, 95% confidence interval (CI) = 1.27–10.60) significantly predicted successful smoking cessation (Nagelkerke *R*^2^ = 0.160, Table 4). When we added the interaction between coach intervention and the use of varenicline, participants in the intervention group who used varenicline had a higher odds ratio of smoking cessation (OR = 9.51, CI = 1.78–50.73). As to smoking reduction, the logistic regression revealed that only the coach intervention (OR = 2.87, 95% CI = 1.06–7.80) significantly predicted successful smoking reduction (Nagelkerke *R*^2^ = 0.093). When we added the interaction between coach intervention and the use of varenicline, participants in the intervention group who used varenicline had a significantly higher odds ratio of smoking reduction (OR = 6.86, CI = 1.35–34.79).

## 4. Discussion

Although this study was suspended from May 2021 due to COVID-19 restrictions for epidemic prevention and drug recall in Taiwan, which impeded us from recruiting the expected number of subjects and caused a smaller sample size, this study still found that the health coaching intervention seemed to enhance the effectiveness of smoking cessation in diabetic patients, especially those who participated in the pharmacotherapy plan. Patients in the intervention group showed a significant smoking reduction, and, even if they did not participate in the pharmacotherapy plan, health coaching seemed to help the patients to reduce smoking effectively; this seems to be consistent with the current rather limited study evidence [22].

In the multivariate analysis, it appears that coaching intervention did increase the effectiveness of using varenicline for smoking cessation. Although the 95% CI was larger due to the small sample size, it seems that exploring the use of both coaching to assist in smoking cessation and varenicline is an interesting research topic. In the past, although there were no studies on the use of health coaches to enhance varenicline or nicotine replacement therapy (NRT), some studies tried to use MI for testing [25,26], and it seems that MI can indeed be effectively paired with NRT or varenicline treatment. However, these studies are also limited by the number of samples or the study design, as in this study, which makes it difficult to directly compare research evidence. In addition to the research design, the nausea caused by the side-effects of varenicline may also be the reason why patients are unwilling to participate in smoking cessation or stop using drugs [25,27], and this was one of the main reasons why the patients in this study stopped taking medication and failed to quit smoking. In addition, as the subjects in this study were limited to patients with type 2 diabetes and smoking, it not only confined this study to a smaller population, but also made it unsuitable for direct comparison with most smoking cessation health coaching studies, because most of these studies are currently aimed at patients with COPD [28,29]. However, we still believe that it is indeed worthwhile to further investigate the effectiveness of using health coaching intervention to enhance the efficacy of NRT or varenicline treatment with larger trials and a more diverse patient population in the future.

The smoking cessation rate in the intervention group was about 48.6% in this study, which is about twice the reported smoking cessation rate of smoking cessation services in Taiwan (27.1% in 2020) [18]; furthermore, in total, 52.9% of the participants reduced their daily number of cigarettes, more than 50% from the baseline. The smoking cessation rates seemed to vary considerably between different studies, and, because of the high heterogeneity of the studies, direct comparisons are not feasible [7,8]. Until now, there are still no conclusions as to which elements of behavioral counseling can effectively increase the smoking cessation rate. For example, one study reported that MI’s decision-making balance method strengthens smokers’ perception of the benefits of smoking [30], while another study suggested that MI may be more suitable for patients with higher education levels, while those with a lower education level are more suitable for five Rs health education [31]. Considering that the five Rs already represent Taiwan’s basic element of smoking cessation education, it was impossible for us to compare the effectiveness of the five Rs with MI or health coaching, such as this study. In addition, as MI and health coaching are not exactly the same methods, we can only conservatively assume that health coaching might be an effective method of smoking cessation. Therefore, more high-quality research is still needed.

There are some hospitals in Taiwan trying to provide MI element health education or conduct related studies [32]; however, until now, MI has not been officially included in Taiwan’s smoking cessation professional training. In addition, since currently Taiwan does not have rigorously certified and supervised MI trainers, and even the use of the five Rs strategy lacks fidelity checks, we think this may indeed affect the effectiveness of smoking cessation in Taiwan. In fact, the fidelity of such behavioral counseling will greatly affect the effectiveness of smoking cessation [33], and this has still been the main limitation of most MI and health coaching studies so far. In contrast, due to the fact that our health coach paid special attention to patient-centered communication methods, adopted positive psychology techniques, and applied his rigorous behavioral coaching training to use MI and other skills proficiently, it allowed our patients to develop deep trust in the medical–patient relationship that should be established by physicians to a certain extent. Indeed, the degree of mutual trust between doctors and patients and the communication skills of medical staff seem to be crucial factors contributing to smoking cessation [34]. This is relatively difficult in the medical environment of Taiwan, because Taiwan’s medical system usually allots less than 5 min of a physician’s time to each patient during a consultation, which is completely insufficient for adequate, in-depth communication and discussion [35]. In particular, application of positive psychology communication skills seems to significantly improve a patient’s intrinsic motivation [36]; however, these skills have not been emphasized enough.

Our study did not find significant impacts of demographic characteristics, smoking experience, or smoking cessation experience on smoking cessation rates, but we found that the number of daily cigarettes did significantly affect the success rate of smoking cessation, which aligns with some study evidence [37]. If there is a lower amount of smoking and a lower degree of addiction, it seems that it is indeed easier to quit smoking successfully. Race may also be an important factor affecting the success rate of smoking cessation [38], but this is not applicable in Taiwan. Different types of jobs and specific industries may also be contributing factors influencing smoking rates, but it is also relatively hard to find significant differences with a small sample size. One study conducted a long time ago saw a significantly higher smoking rate among workers of material-moving occupations, construction laborers, and vehicle mechanics and repairers [39]. In this study, our health educators and the coach also found such a tendency. The reason why many participants were unable to completely quit smoking seemed to be because almost everyone in their workplace smoked, which made it difficult for them to resist the temptation of cigarettes, and the pressure of rapport or human relationship also made it more difficult to quit smoking. Such effects and the addictive nature of nicotine may make it difficult for pure behavioral counseling to work. When these effects are reduced due to restrictions caused by the COVID-19 pandemic, the smoking cessation rate would naturally increase [40]. We decided not to continue the original research plan in view of the fact that we could not effectively judge the impact of the COVID restrictions on the smoking cessation rate. In summary, determining how to help patients reduce the impact of the workplace environment and human pressure on smoking cessation is still a very important research topic in the future.

On the basis of our findings, health coaching intervention in smoking cessation seemed to have significantly higher quit rates when combined with a pharmacotherapy plan. Given that smoking cessation is a complex behavior involving lifestyle changes, addiction symptoms, and side-effects, we do not consider health coaching to be ineffective per se, but rather an effective element of a smoking cessation program that can bolster an original medication program. Hence, the study proposes some suggestions for future studies and our medical system. Firstly, more full-time professional behavior coaching services need to be introduced in Taiwan. The current medical system greatly limits the time and effectiveness of communication between doctors and patients, and there is also a lack of adequate smoking cessation behavior counselors with rigorous training and regular fidelity supervision. Secondly, more studies on health coaching with high-quality evidence are needed. Although this study seemed to have acceptable results, the number of samples did not meet the recruiting expectations, and the research was interrupted by the restrictions which could have affected the effectiveness of the evidence in this study.

Although this study was not completed as scheduled, it still had some strengths. Firstly, the original randomized controlled design allowed this study to retain a certain degree of evidence, even if it was affected by some problems as described above, and smoking cessation studies specifically for diabetic patients are also quite rare. Secondly, the use of rigorously trained and certified coaches to perform the smoking cessation counseling was also one of the strengths in this research, as it may have greatly improved the quality of coaching. The training and practice fidelity of the coaches remain important factors influencing research in such behavioral interventions [41,42]. The small sample size was the main limitation of this study. The main reason might be that this study targeted patients with type 2 diabetes, whereby only about 10% of T2DM patients smoked, which limited the sample characteristics of this study. Secondly, another important limitation is that biochemical tests such as carbon monoxide measurements or urine cotinine tests were not used; instead, only FTND scores were used considering COVID-19 prevention during the pandemic, which might have overestimated the effect of smoking cessation. The reason for this design is mainly because that the current referral standard for smoking cessation plans in Taiwan is based on FTND scores; however, given that some studies have found that tests such as CO testing may increase the motivation of patients to quit smoking, this may be one of the standardized procedures that the Taiwan HPA can incorporate into routine smoking cessation plans in the future [22]. It is suggested that similar studies with larger sample sizes, longer follow-up times, and more objective indicators of smoking cessation should be carried out to confirm the validity of the study’s findings in the post-COVID-19 era.

## 5. Conclusions

The study found that health coaching may improve the effectiveness of tobacco control project in Taiwan with a reduction in the number of cigarettes smoked and increase the cessation rate of type 2 diabetic patients who participate in the pharmacotherapy plan. More prospective health coaching studies related to smoking cessation are needed before implementing health coaching services into smoking cessation plans in Taiwan.

## Figures and Tables

**Figure 1 ijerph-20-04994-f001:**
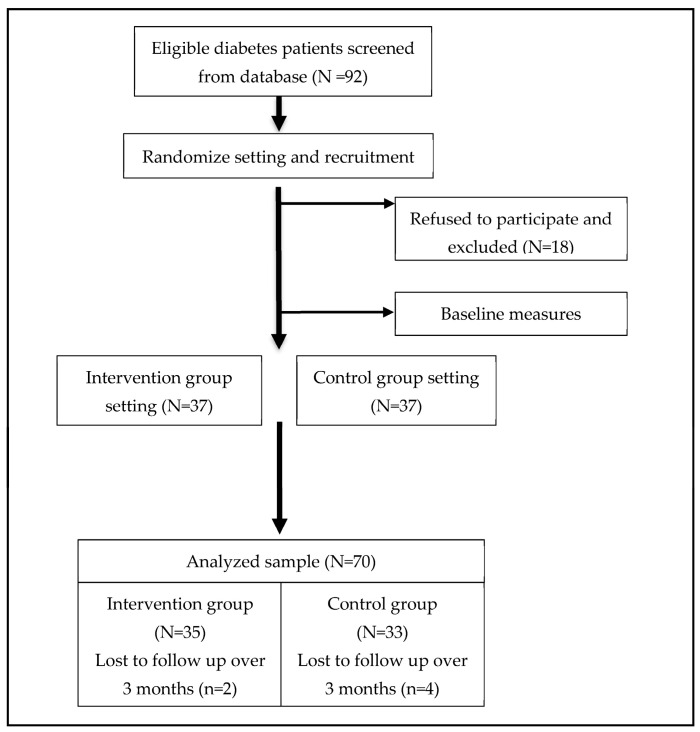
Flow diagram of participants: recruitment, intervention and follow-up.

**Table 1 ijerph-20-04994-t001:** Demographic characteristics and baseline values of the study groups.

	Demographic Characteristics *N* (%)		*p*-Value
	Intervention Group (*n* = 35)	Control Group (*n* = 33)	
Gender			0.600
Male	28 (80.0)	28 (84.8)	
Female	7 (20.0)	5 (15.2)	
Educational level			0.861
Senior high school and below	20 (57.2)	21 (63.6)	
University and above	15 (42.9)	12 (36.4)	
Experience with smoking cessation			0.454
Previously tried to quit	5 (14.3)	7 (21.2)	
Never tried to quit	30 (85.7)	26 (78.8)	
Age, years (mean ± SD)	57.23 ± 9.67	54.70 ± 11.55	0.330
Number of daily cigarettes	20.40 ± 10.50	16.82 ± 10.16	0.158
Length of smoking (years)	35.97 ± 7.99	33.03 ± 11.16	0.214
Addicted to nicotine ^a^	5.57 ± 2.03	5.03 ± 2.85	0.368

^a^ Fagerström Test for Nicotine Dependence (FTND).

**Table 2 ijerph-20-04994-t002:** Result of smoking cessation between study groups.

	Result of Smoking Cessation *N* (%)		*p*-Value
	Intervention Group (*n* = 35)	Control Group (*n* = 33)	
Smoking cessation (FTND = 0 point)			0.309
Yes	17 (48.6)	12 (36.4)	
No	18 (51.4)	21 (63.6)	
Smoking reduction			0.030 *
≥50%	23 (65.7)	13 (39.4)	
<50%	12 (34.3)	20 (60.6)	
Reduced cigarettes (mean ± SD)	12.88 ± 9.28	7.74 ± 10.03	0.032 *

* 0.01 < *p* < 0.05.

**Table 3 ijerph-20-04994-t003:** Result of smoking cessation and smoking reduction of the groups.

	With the Use of Varenicline		
	Intervention Group	Control Group	*p*-Value
Smoking cessation (FTND = 0 points)			0.082 ^#^
Yes	12 (70.6)	6 (40.0)	
No	5 (29.4)	9 (60.0)	
Smoking reduction			0.502
≥50%	9 (50.0)	7 (38.9)	
<50%	9 (50.0)	11(61.1)	
	Without using varenicline		
Smoking cessation (FTND = 0 point)			0.717
Yes	5 (27.8)	6 (33.3)	
No	13 (72.2)	12 (66.7)	
Smoking reduction			0.014 *^,a^
≥50%	14 (82.4)	6 (40.0)	
<50%	3 (17.6)	9 (60.0)	

* 0.01 < *p* < 0.05, ^#^ 0.05 < *p* < 0.10. ^a^ Analyzed by Fisher’s exact test.

**Table 4 ijerph-20-04994-t004:** Result of logistic regression.

	Smoking Cessation OR (95% CI)		Smoking Reduction OR (95% CI)	
	Full Group (*n* = 68)	Coaching Intervention Only (*n* = 35)	Full Group (*n* = 68)	Use Varenicline Only (*n* = 32)
Varenicline use	3.67 * (1.27–10.60)	9.51 ** (1.78–50.73)		
Coaching intervention			2.87 * (1.06–7.80)	6.86 * (1.35–34.79)
Number of daily cigarettes	0.95 (0.90–1.00)	0.91 (0.83–1.00)	1.00 (0.96–1.06)	1.01 (0.94–1.10)
Nagelkerke *R*^2^	0.160	0.384	0.093	0.247

* 0.01 < *p* < 0.05, ** *p* < 0.01.

## Data Availability

The data presented in this study are available on request from the corresponding author. The data are not publicly available due to the Institutional Review Board of Cathay General Hospital privacy protection policy.

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
