# Peer review of "Effectiveness of Health Coaching in Smoking Cessation and Promoting the Use of Oral Smoking Cessation Drugs in Patients with Type 2 Diabetes: A Randomized Controlled Trial"

_ijerph, 2023, doi:10.3390/ijerph20064994_

Round 1

Reviewer 1 Report

- While writing smoking-cessation, should a hyphen be used in the middle? Standardize it.

- While mentioning from health coaching intervention in abstract, please detail it more.

- The last sentence of abstract is unnecessary.

- Get shorten the introduction.

- In pharmacotherapy plan, why only varenicline? NRT, bupropion, etc.. !!

- Number of participants is less.

- 80% power is less.

- CO measurement is not used also for assessing the level of addiction, please do not use that part.

- Number of control group should not be less than intervention group.

- References are old and should be updated.

Author Response

1.  While writing smoking-cessation, should a hyphen be used in the middle? Standardize it.

We’ve standardized it, no hyphen in the middle.

2.  While mentioning from health coaching intervention in abstract, please detail it more.

We have added a description based on your suggestion, and considering the word limit of the abstract, we think such a change should be appropriate.

3. The last sentence of abstract is unnecessary.

It is deleted it now.

4. Get shorten the introduction.

Since the other two reviewers think that the introduction has appropriate content and do not suggest major revision to it, we have decided not to make any changes or shorten it.

5. In pharmacotherapy plan, why only varenicline? NRT, bupropion, etc.. !!

As mentioned in the method, the Taiwan government’s drug treatment plan at the time of study was mainly based on varenicline. In accordance with the government's policy promotion, the Cathay general hospital also used varenicline as the main treatment plan. However, following a drug safety incident, the original varenicline treatment was replaced by NRT and other programs.

6. Number of participants is less.

This is due to the small sample size of the population. As mentioned in the introduction, the smoking rate among adults in Taiwan is about 13.1%, with higher rates among young and middle-aged people. The study focused on patients with Type 2 diabetes who smoked, which accounted for only about 10% of the total T2DM patients, and thus limited the number of recruits.

7. 80% power is less.

This is because 80% power is commonly used in most health coaching studies when calculating the sample size. In order to ensure the comparability of our study with other studies, we also use the 80% power standard.

8. CO measurement is not used also for assessing the level of addiction, please do not use that part.

We did not use the CO test in this study due to the COVID-19 epidemic. However, traditionally most studies related to smoking cessation use the test of concentration of blood CO as one of the indicators to determine whether a patient has smoked during the assessment period, so we think we should explain why the indicator was not used.

9. Number of control group should not be less than intervention group.

This is because both groups had a small number of withdrawals after the intervention. As shown in Figure 1, the original recruitment of the two groups had the same number.

10. References are old and should be updated.

We are not sure which references you are referring to, so we can only update some data such as prevalence rate in the introduction. We believe that the references we have cited are the most appropriate.

Reviewer 2 Report

Dear authors, 

congratulations on your research and on a well-written paper. 

I find you paper to be generally very well structured, the research design scientifically correct, and I overall like you study and results.

I have a few observations which I think would improve your paper:

1) You mention that the randomization was done via a computer software. Please specify what randomizer software you used, in order to improve transparency of the research.

2) You discuss in detail the limitations and issues you encountered during the study, which I really like. Do you think that knowing they participate in study might have influenced the coach's "performance" or made them try harder to obtain the desired results with their work group? If you had any discussions with them regarding their behavior, please detail in the manuscript. I think this question is important to assess if the results are for an "average" coaghing program, and not special "try extra hard" coaching. Was there only one coach, or a team of different coaches?

3) There are some minor spelling errors, please do another proofread.

I think your article can be published with these, and some other small improvements (recommended by the other reviewers)

Author Response

1) You mention that the randomization was done via a computer software. Please specify what randomizer software you used, in order to improve transparency of the research.

We used PASW version 22.0 for random assignment, which provides this functionality.

2) You discuss in detail the limitations and issues you encountered during the study, which I really like. Do you think that knowing they participate in study might have influenced the coach's "performance" or made them try harder to obtain the desired results with their work group? If you had any discussions with them regarding their behavior, please detail in the manuscript. I think this question is important to assess if the results are for an "average" coaghing program, and not special "try extra hard" coaching. Was there only one coach, or a team of different coaches?

There was only one health coach in this study, and, to our knowledge, there is only one ICF-certified health coach in Taiwan.

You raised an interesting question that sounds a bit like the Hawthorne effect, which we'd be more than happy to answer here, but it doesn’t seem suitable to add to the Discussion paragraph after we think about this carefully, so we decide to answer it here.

About this question, we believe it is related to a problem that all interventional studies may encounter, which is whether the intervention effect fades after the intervention ends. It is possible that participants may be more conscientious about implementing a specific intervention during the study because they know they are participating in a program and are being supervised by the investigator. For this reason, some researchers have specifically looked at whether the effects of interventions wear off within one to two years after the program ends.

For example, some studies have specifically designed an additional half-year to one-year follow-up window to explore the degree of behavior extinction, and they have found that the effects of behavioral interventions will indeed gradually fade after the intervention ends. So, some supplementary counseling may be required on a regular basis[1].

In this regard, we believe that this is not only human nature, but also quite understandable. Because of this, health coaching focuses more on the adult learning to enable patients to acquire a new habit. Therefore, when they receive supplementary coaching again, instead of having to relearn it, they can recover fast by evoking specific experiences from previous coaching.

That's why in the Discussion we discuss how to integrate effective behavioral coaching into routine medical care rather than treating it as a single independent service.

Since this study did not include a follow-up period long enough to explore the possible waning of coaching effectiveness (such as starting to smoke again), we think that the above discussion may not be suitable to be added to the existing Discussion.

  1. Sharma, A.E., et al., What Happens After Health Coaching? Observational Study 1 Year Following a Randomized Controlled Trial. Annals of Family Medicine, 2016. 14(3): p. 200-207.

3) There are some minor spelling errors, please do another proofread.

I think your article can be published with these, and some other small improvements (recommended by the other reviewers)

Reviewer 3 Report

I have received for review an original research article entitled "Effectiveness of health coaching in smoking cessation and pro-2 moting the use of oral smoking-cessation drugs in patients 3 with type 2 diabetes: A Randomized-Controlled Trial" which is being processed for publication in the journal International Journal of Environmental Research and Public Health.

The authors have proposed a manuscript with an interesting theme with important medical and socio-economic implications. Smoking cessation can be considered a primary prevention strategy with demographic implications over time by reducing the risk of developing diabetes or cardiovascular disease.

Introduction: It contains relevant, up-to-date information for the country in which the study was conducted.

Materials and methods:

1.       I recommend the authors to mention the ethical aspects at the end of the section in a special subsection containing international ethical benchmarks (e.g. Declaration of Helsinki).

2.       The authors refer to a figure 1 that is not in the manuscript.

3.       Line 146 – remove the word “the” which is strikethrough

4.       Line 153 – define MI

5.       Mention which variables were analyzed (demographics, comorbidities, etc...)

Results

6.       Lines 236-240 - These data should be found in the materials and methods section, not under results. Explain how the 92 patients were selected from the database. It is difficult to believe that in such a long period of time there are so few diabetic patients. Also, patient selection must be continuous.

7.       Line 240 – no figure 1 included in the manuscript

8.       Lines 277-288 - Add a table with the logistic regression performed including the p-value

Discussions: Well structured, in accordance with the literature and in relation to local medical data.

References: Please insert the references in accordance to the journal Reference Guidelines. In the text, reference numbers should be placed in square brackets [ ], and placed before the punctuation;

Author Response

Introduction: It contains relevant, up-to-date information for the country in which the study was conducted.

Materials and methods:

1. I recommend the authors to mention the ethical aspects at the end of the section in a special subsection containing international ethical benchmarks (e.g. Declaration of Helsinki).

We’ve added it at Line 105.

2. The authors refer to a figure 1 that is not in the manuscript.

We’ve added figure 1 into the manuscript now.

3. Line 146 – remove the word “the” which is strikethrough

It is deleted now.

4. Line 153 – define MI

Considering the several different skills and theories mentioned in this paragraph, it seems that it is not suitable to explain or define one of them separately. In addition, after several attempts to modify the content, we think it is really difficult to specifically explain what MI is in this section, so we have decided not to adjust that paragraph in method, but added some sentences in introduction (Line 80).

5. Mention which variables were analyzed (demographics, comorbidities, etc...)

 We don't quite understand which part you are referring to. We reviewed the descriptions of 2.6 and 2.7 and found that the description of the variables is sufficient.

Results

6. Lines 236-240 - These data should be found in the materials and methods section, not under results. Explain how the 92 patients were selected from the database. It is difficult to believe that in such a long period of time there are so few diabetic patients. Also, patient selection must be continuous.

We have moved it to Line 129 of method based on your suggestion

Our potential subjects were patients with T2DM and still smoking. Considering the adult smoking rate in Taiwan is about 13.1%, and it is higher among young and middle-aged people, we think it is reasonable that we have a small population for recruitment. As described in the method, we first screened out eligible patients (who have T2DM and are still smoking) from the database of the Endocrinology Department of the Cathay general hospital and then recruited them.

7. Line 240 – no figure 1 included in the manuscript

We’ve added figure 1 into the manuscript now.

8. Lines 277-288 - Add a table with the logistic regression performed including the p-value

 We have added table 4 according to your suggestion, with * for p value

Discussions: Well structured, in accordance with the literature and in relation to local medical data.

References: Please insert the references in accordance to the journal Reference Guidelines. In the text, reference numbers should be placed in square brackets [ ], and placed before the punctuation;

We have corrected it now.

Round 2

Reviewer 3 Report

Thanks to the authors for the changes made.

Figure 1 should be included in the materials and methods section at 2.1. study design.

Author Response

We have moved Figure 1 to "2.1 Study procedures and randomization settings" based on your suggestion (line 127)